# Dynamics of recombination, X inactivation and centromere proteins during stick insect spermatogenesis

Zoé Dumas[ID][1☯*], William Toubiana[ID][1☯*], Marie Delattre[ID][2], Tanja Schwander[ID][1]

**1** Department of Ecology and Evolution, University of Lausanne, Lausanne, Switzerland, **2** Laboratory of Biology and Modeling of the Cell, Ecole Normale Supérieure de Lyon, CNRS UMR 5239, Inserm U1293, Université Claude Bernard Lyon 1, Lyon, France

☯ These authors contributed equally to this work.
* william.toubiana@unil.ch (WT); zoe.dumas@unil.ch (ZD)

## Abstract

In eukaryotes, the cellular processes contributing to gamete formation form the building blocks of genetic inheritance across generations. While traditionally viewed as conserved across model organisms, emerging studies reveal significant variation in meiotic and post-meiotic processes. Extending our knowledge to non-model organisms is therefore critical to improve our understanding of the evolutionary origin and significance of modifications associated with gamete formation. We describe the cytological patterns underlying chromosome segregation, recombination, and meiotic sex chromosome inactivation during male meiosis in the stick insect group *Timema*. Our results provide a detailed description of centromere protein loading dynamics during spermatogenesis, and further reveal that 1) recombination initiates before synapsis (unlike *Drosophila* meiosis), and 2) that the X remains actively silenced despite two waves of transcriptional activation in autosomes during spermatogenesis. Together, our observations help understand the evolutionary significance of key cellular events related to spermatogenesis and shed light on the diversity of their associated molecular processes among species, including *Timema* stick insects.

## Author summary

To ensure the faithful transmission of genetic material between parents and offspring, chromosomes undergo a series of coordinated processes prior to fertilization that include pairing with the homologous partner, exchanging pieces of DNA (i.e., recombination), segregating (i.e., separating), and remodeling the compaction and transcription of DNA. In males, these cellular processes are divided into two main phases: *meiosis*, where the genetic material is equally split into four gametic cells, and *spermiogenesis*, where chromosomes are tightly packed into the compact heads of sperm cells. Although these processes are

**Data availability statement:** All data needed to replicate the results and evaluate the conclusions in the paper are present in the paper and/or the Supplementary Materials.

**Funding:** European Research Council Consolidator Grant (No Sex No Conflict to T.S., URL: [https://erc.europa.eu)]https://erc.europa.eu) and Fonds National Suisse grant 31003A_182495 (to T.S., URL: [https://www.snf.ch)]https://www.snf.ch). W.T. received a salary from European Research Council Consolidator Grant (No Sex No Conflict to T.S., URL: [https://erc.europa.eu)]https://erc.europa.eu). The funders had no role in study design, data collection and analysis, decision to publish, or preparation of the manuscript.

**Competing interests:** The authors have declared that no competing interests exist.

crucial for reproduction across all animals, the underlying mechanisms can vary widely between species. In this study, we explore how these processes take place in *Timema*, a group of stick insects. We reveal that while the mechanisms, and their temporal relationships, related to chromosome pairing and recombination are similar to other organisms, the mechanisms involved in chromosome segregation and transcription show notable differences.

## Introduction

A key step in sexual reproduction is the generation of haploid cells. The transition from diploidy to haploidy occurs in meiosis, a cellular process that ensures the maintenance of ploidy levels between generations. During meiosis, chromosomes are inherited from a diploid progenitor germ cell to produce four haploid gametic cells, through two rounds of cell division [1]. In males, these events are followed by the process of spermiogenesis, during which the four meiotic products mature into sperm cells which then fuse with female gametes and restore the diploid cellular state [2]. While traditionally viewed as conserved across model organisms, emerging studies reveal significant variation in meiotic and post-meiotic processes underlying sperm maturation [2–6].

In a canonical depiction of meiosis, the early steps (i.e., prophase I) involve chromosome condensation, pairing, synapsis, and recombination [1]. These processes work together to facilitate the segregation of chromosomes and formation of novel allele combinations [1]. However, the temporal and functional relationships between these steps can vary among species. For example, some lineages exhibit achiasmatic meiosis, where homologous chromosomes segregate without recombination, as is the case for *Drosophila* fruit fly males [7]. In organisms that do recombine, the temporal relationships between meiotic recombination and synapsis can also vary, with dependence or independence of synapsis on the occurrence of double-stranded breaks (DSBs) and recombination [8]. Although recombination plays a key role in homologous chromosome segregation, it appears that different features can coexist. Yet, the evolutionary origins and causes of this variation remain unclear [7,9].

Another variable phenotype is observed during the early steps of meiosis and in species with differentiated sex chromosomes such as in XY or X0 systems (i.e., without Y chromosome). Here, sex chromosomes partially synapse or do not synapse at all [10]. This phenomenon is thought to trigger major transcriptional changes, by which sex chromosomes are transcriptionally silenced during meiosis through a process known as meiotic sex chromosome inactivation (MSCI) [11]. However, there is extensive variation in these transcriptional changes among species, with MSCI occurring in some species but not others [12–16]. Additionally, the timing and mechanisms of MSCI differ between species, preventing the formulation of a general theory as to why MSCI occurred in the first place [11,17] (see the discussion section for a list of proposed hypotheses).

Following prophase-I events, two rounds of cell division ensure the proper segregation of each homologous chromosome and chromatid. This process is mediated by centromeres—specific chromosomal regions where centromere proteins bind to DNA and facilitate the attachment of the spindle microtubules [18]. However, the way centromeres are specified varies considerably among species [19]. In many eukaryotes, centromeres are confined to specific regions of the chromosomes and are marked by the histone variant CenH3 (also known as CENP-A), which recruits other centromere proteins before cell division [20]. These organisms, known as monocentric species, contrast with the holocentric species, where centromeres are specified along the entire length of chromosomes [21]. Furthermore, the loading of centromere proteins, including CenH3, shows considerable diversity; they can be deposited through various mechanisms, and in some species, they are even entirely absent [19,22]. How such changes can evolve and how the cellular machinery adapts to ensure accurate chromosome segregation remains poorly understood.

Finally, the process of sperm maturation (i.e., spermiogenesis) involves a series of chromosomal regulation that include changes in transcription and DNA compaction. While these events are essential for the transmission of genetic information across generations, they also pose significant challenges for preserving genome integrity. For example, transcription late during spermatogenesis or the replacement of canonical by specialized histones exposes DNA to increased risks of damage that cannot be repaired in haploid cells [23]. Whether sexual organisms address these challenges in similar or different manners remains largely unclear.

Understanding the diversity of meiotic and post-meiotic processes has become a critical concern for identifying the elements essential for gamete production and elucidating the proximate and ultimate mechanisms underlying the diversification of gametogenesis.

The stick insect genus *Timema* has emerged as a compelling system for exploring the diversity of cellular processes linked to spermatogenesis (i.e., male meiotic and post-meiotic processes), owing to two recently uncovered features during male meiosis [24,25]. First, the centromere histone determinant CenH3 binds in a localized or chromosome-wide manner depending on chromosomes and meiotic stages [25]. Second, the X chromosome demonstrated significant alterations in transcriptional activity over gonad development, diminishing in overall gene expression as the meiotic program initiates [24]. These findings prompted us to investigate in more detail the cellular processes involved in *Timema* male meiosis. While our main focus was on the species *Timema californium*, we also used a comparative approach across different *Timema* species, covering over 30 MY of divergence [26], and compared our findings with the cellular processes described in other organisms (see Materials and Methods for the details and limitations in the species comparisons). This approach allowed us to explore the diversity and spatiotemporal dynamics of different cellular processes involved in recombination, meiotic sex chromosome inactivation and chromosome segregation. Overall, our study unveils that despite a conservation in the cytological processes governing spermatogenesis in *Timema*, the spatiotemporal dynamics of the molecular machinery involved can vary considerably among species, particularly when compared to model organisms.

## Results

### *Timema* chromosomes pair and initiate recombination before synapsis during male prophase I

Meiotic recombination is initiated by the formation of programmed DNA double-stranded breaks (DSBs) that rapidly induce the phosphorylation of the histone variant H2AX (γH2AX) and the recruitment of the recombinase Rad51 that is involved in homologous recombination and mediates the single-strand invasion to repair DNA after DSBs [27,28]. To determine whether DSB formation and the initiation of homologous recombination are required for initiating synapsis during *Timema* male meiosis, which is chiasmatic [29], we analysed the temporal localisation of these two proteins, as well as the subunit of the cohesin complex SMC3. SMC3 marks chromosomal axes but can also be used as marker for the progression of synapsis as SMC3 filaments become thicker as chromosomes pair and synapse [30]. We find that the induction of recombination, including DSB formation and DNA repair via strand invasion, starts prior to synapsis, when chromosomes begin

to condense and elongate into strands (see materials and methods for the characterization of cellular stages). Indeed, during the early zygotene stage, γH2AX and Rad51 form a cloud of foci at a polarized region of the nucleus (Figs 1A–1E, 2A–2E and Figs A panels A-J and B panels A-O in S1 Text). At the same stage, only thin continuous SMC3 filaments are observed (Figs 1A–1C, 2A–2C and Fig C in S1 Text), forming a structure commonly referred to as the bouquet configuration in other species [31,32]. The polarized patterns of γH2AX and Rad51 proteins as well as the bouquet configuration were also observed in grasshoppers and a true bug's spermatocytes [9,33], suggesting that the process is conserved across insect orders from the polyneoptera and condylognatha clades.

During the mid-zygotene to pachytene transition, synapsis progresses gradually and asynchronously among chromosomes, alongside with the resolution of the induced DSBs. This was evidenced by SMC3 filaments gradually shifting from thin to thicker filaments, with a variable number of thick filaments among cells, while the γH2AX signal progressively homogenizes and decreases over the entire nuclear surface (Fig 1F–1O and Figs A panels A-O and C panels E-J in S1 Text). Similarly, the Rad51 signal decreases and only a subset of recombination sites per chromosome persists as discrete foci along the thick SMC3 filaments (Fig 2F–2T and Figs B panels A-E and B panels P-T in S1 Text).

Before the initiation of the first meiotic division, γH2AX and Rad51 signals disappear while SMC3 filaments become interrupted near chromosomal ends (Fig C panels O-P and C panels A'-L' in S1 Text). These interruptions likely highlight chiasma sites that correspond to the contact zone where homologous chromosomes remain connected [34]. At metaphase I, chromosomes then form "rod" bivalents with cross-shaped SMC3 labeling interrupted at one chromosomal end for the majority of chromosomes in all four *Timema* species examined, suggesting the formation of a single terminal or sub-terminal chiasma site (Fig C panels O-P and C panels A'-L' in S1 Text). We also identified two *Timema* species often forming a "ring" bivalent on one chromosome (cells without "ring" bivalents do also exist but represent a minority) with more circularized SMC3 labeling interrupted at two chromosomal ends, suggestive of two terminal or sub-terminal chiasma sites (Fig C panels O-P and C panels A'-F' in S1 Text).

Overall, our findings show that the phosphorylation of the histone H2AX and the chromosomal deposition of the recombinase Rad51 occur prior to the thickening of SMC3 filaments, supporting the idea of a recombination-dependent machinery for synapsis formation as observed in *Arabidopsis,* yeast, jellyfish and mouse [35–38]. Furthermore, the terminal positions of chiasma sites suggest that crossing-overs may preferentially occur on chromosomal ends in *Timema* males, although such conclusion requires further molecular evidence.

### Spermatogenesis progression of meiotic sex chromosome inactivation in *Timema* males

In *Timema* male meiosis, the X chromosome remains unsynapsed due to the absence of a homologous partner (i.e., X0 system) [39]. As a result, the pachytene stage is marked by thick SMC3 filaments on all chromosomes except the X chromosome, where a thin SMC3 filament remains (Fig C panel I in S1 Text) (see materials and methods for X vs autosome identification). At the same stage, we observed that the X chromosome is also characterized by an intense DAPI staining, not present at the zygotene stage (Fig C panels C-J in S1 Text). This is reminiscent of the heteropycnotic body (also termed sex body) found in organisms with meiotic sex chromosome inactivation (MSCI) [16,40]. The heteropycnotic body is a key marker of MSCI, highlighting the intense compaction of sex chromosomes, and is often assisted by the DNA damage response (DDR) pathway in mammals, where diverse DDR factors including γH2AX and Rad-related kinases accumulate to initiate the chromosome-wide silencing [15,41]. However, this pathway appears to be modified in hemipteran (XY system) and orthopteran (XO system) insects, notably with the absence of Rad51 and the emergence of γH2AX only after the sex body has formed [16,42,43]. Our analyses of γH2AX and Rad51 in *Timema* suggest that this cascade is also modified compared to other insects and mammals. Indeed, despite the detection of Rad51 proteins at discrete sites along the unsynapsed axial element of the X chromosome (Fig 2P–2T and Fig B panels A-E and B panels P-T in S1 Text), the γH2AX signal is inconsistently detected across *Timema* species during pachytene (Fig 1K–1O and Fig A panels A-O in S1 Text). Specifically, *T. californicum* and *T. petita* exhibit two to three distinct γH2AX patches in X regions, while no signal

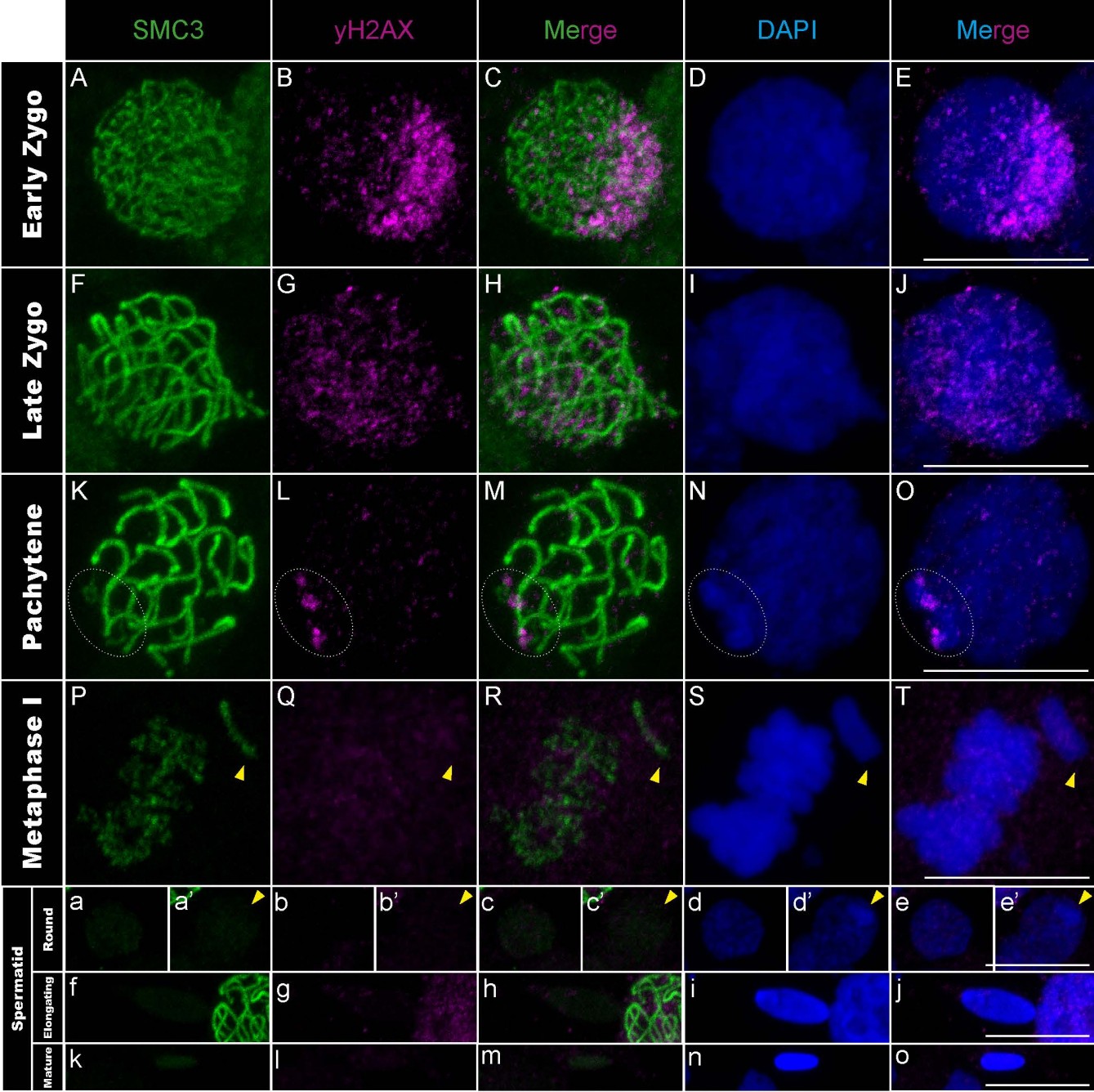

**Fig 1. γH2AX distribution in *T. californicum* meiocytes and spermatids.** Projections of stack images acquired across *T. californicum* squashed meiocytes (A–T) and spermatids (a–o), at the stages indicated, stained with DAPI (blue) and double immunolabeled for SMC3 (green) and γH2AX (magenta). γH2AX labeling is intense and granular in one nuclear pole, then gradually homogenizes and decreases as synapsis progresses, except on the X chromosome where three patches form in pachytene. The position of the X chromosome is indicated with arrowheads or dashed circles in pachytene (see materials and methods for X vs autosome identification). Scale bar: 10 μm. See Fig A in S1 Text for screens in males of other *Timema* species.

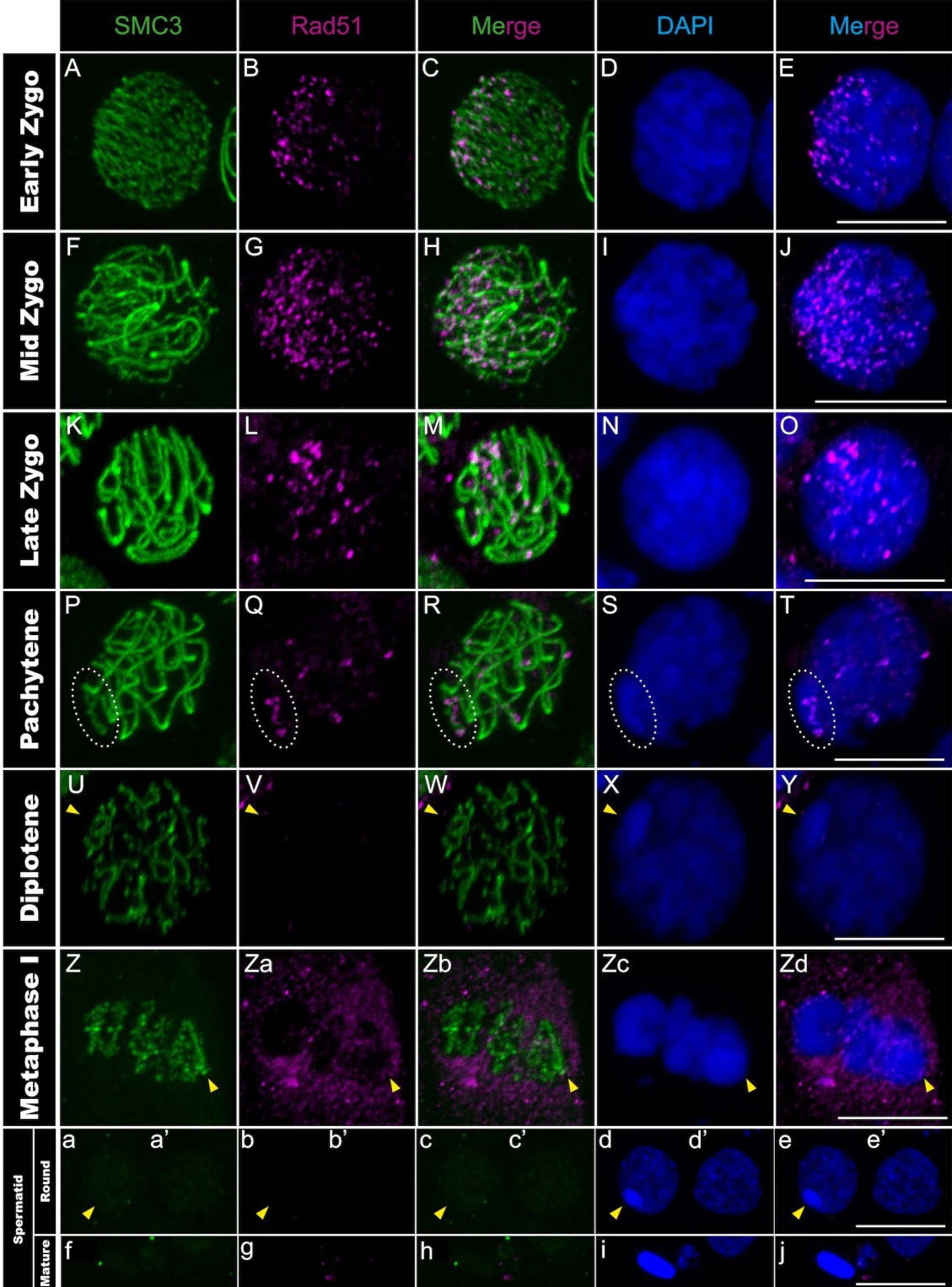

**Fig 2. Rad51 distribution in *T. californicum* meiocytes and spermatids.** Projections of stack images acquired across *T. californicum* squashed meio-cytes (A–Zd) and spermatids (a–j), at the stages indicated, stained with DAPI (blue) and double immunolabeled for SMC3 (green) and Rad51 (magenta). During the early to mid-zygotene transition, there is an increase in the number of Rad51 foci and a localisation in one nuclear pole, as for γH2AX. As

synapsis progresses, the number of Rad51 foci decreases and a few remain on SMC3 axes in pachytene. An accumulation of Rad51 foci on the X chromosome is also observed in pachytene. The position of the X chromosome is indicated with arrowheads and the associated Rad51 signal along its SMC3 axis with dashed circles in pachytene. Scale bar: 10 μm. See Fig B in S1 Text for screens in males of other *Timema* species.

is detected on the X chromosome for *T. cristinae* and *T. chumash*. Note that because we used a commercial antibody against γH2AX, we cannot formally exclude that this variation across species is caused by variable antibody affinities or cross-reaction with other proteins. However, the antibody clearly detects the expected cloud of DSB-associated γH2AX foci at a polarized region of the nucleus during the zygotene stage in all *Timema* species (Fig 1A–1E and Fig A panels A-J in S1 Text). We could also verify that the presence or absence of γH2AX on the X chromosome at pachytene is consistent across individuals of the same species (Table A in S1 Text). Finally, the X chromosome in all *Timema* species exhibits intense DAPI staining at the nuclear periphery, attesting that the unusual γH2AX pattern on the X chromosome, in some species, does not prevent the tight compaction of this chromosome (Fig 1N–1O and Fig A panels A-O in S1 Text).

We previously showed that the X chromosome compaction in pachytene is associated with MSCI [24]. However, it is unknown whether the X chromosome in *Timema* is transcriptionally inactive throughout spermatogenesis or only at specific stages, including the pachytene stage. To determine the transcriptional status of the X chromosome and autosomes, we immuno-labelled the RNA polymerase II phosphorylated on serine 2, which is a conserved transcriptional marker of transcription elongation [44]. We found that the X chromosome remains transcriptionally inactive throughout spermatogenesis, while there are two phases of autosomal transcription during meiosis and spermiogenesis. Indeed, the RNA polymerase II signal is first absent from the entire nucleus at the onset of prophase I (i.e., leptotene and zygotene stages; Fig 3A–3J). It then appears at the pachytene stage only on the autosomes, concomitantly with the formation of the heteropycnotic body (Fig 3K–3O). This first transcriptional phase is maintained until the diplotene stage (Fig 3P–3Zi) and is followed by a second transcriptional phase during the spermatid stage, marked again by an intense RNA polymerase II signal on the autosomes but not the X chromosome (Fig 3a–e'). As spermatids then elongate, the polymerase II signal is depleted from the nuclei and the heteropycnotic body is no longer visible (Fig 3f–3o).

Contrasting with the absence of polymerase II signal, we observed that the X chromosome becomes coated by the modified histone H3K9me3 in pachytene (Fig 4F–4J and Fig D panels A-F and D panels L-P in S1 Text), another MSCI marker involved in transcriptional repression, heterochromatization, and heteropycnotic body formation of the sex chromosomes during MSCI [16,41]. However, while the heteropycnotic body persists in all species until the round spermatid stage, the maintenance of H3K9me3 on the X varies among species: it remains until the round spermatid stage in *T. petita* and *T. chumash* whereas its signal is already lost at this stage in *T. californicum* and *T. cristinae* (Fig 4a–4e' and Fig D panels A-C and D panels G-P in S1 Text). We note that species lacking H3K9me3 signal on the X chromosome at the spermatid stage, display a coating of H3K9me3 on the X chromosome in pachytene and other meiotic stages such as metaphase I (Fig 4F–4T and Fig D panels A-C in S1 Text), indicating that the binding efficiency of the antibody is not affected in these species.

In summary, our findings show the complete transcriptional inactivity of the X chromosome throughout *Timema* spermatogenesis, which explains the strongly reduced overall gene expression of the X chromosome compared to autosomes in male gonads [24]. Furthermore, the establishment and maintenance of this process appear to be dynamic, unfolding across five major cellular steps. During the first step, the polymerase II is absent from nuclei in leptotene and zygotene stages, and all chromosomes, including the X chromosome, are most likely transcriptionally inactive (Fig 3A–3J). The second step involves the appearance of the MSCI markers such as the formation of the heteropycnotic body along with the presence of H3K9me3, γH2AX and Rad51 proteins on the X (Fig 1K–1O and Fig A panels A-O, B panels P-T and B panels A-T in S1 Text). While this process appears to be dependant on H3K9me3 and Rad51 (i.e., consistently detected across *Timema* as in mammals [15]), it does not seem to require the accumulation of γH2AX on the X chromosome (i.e., inconsistently detected across *Timema* and resembling the pattern in grasshoppers but differing from that in mammals

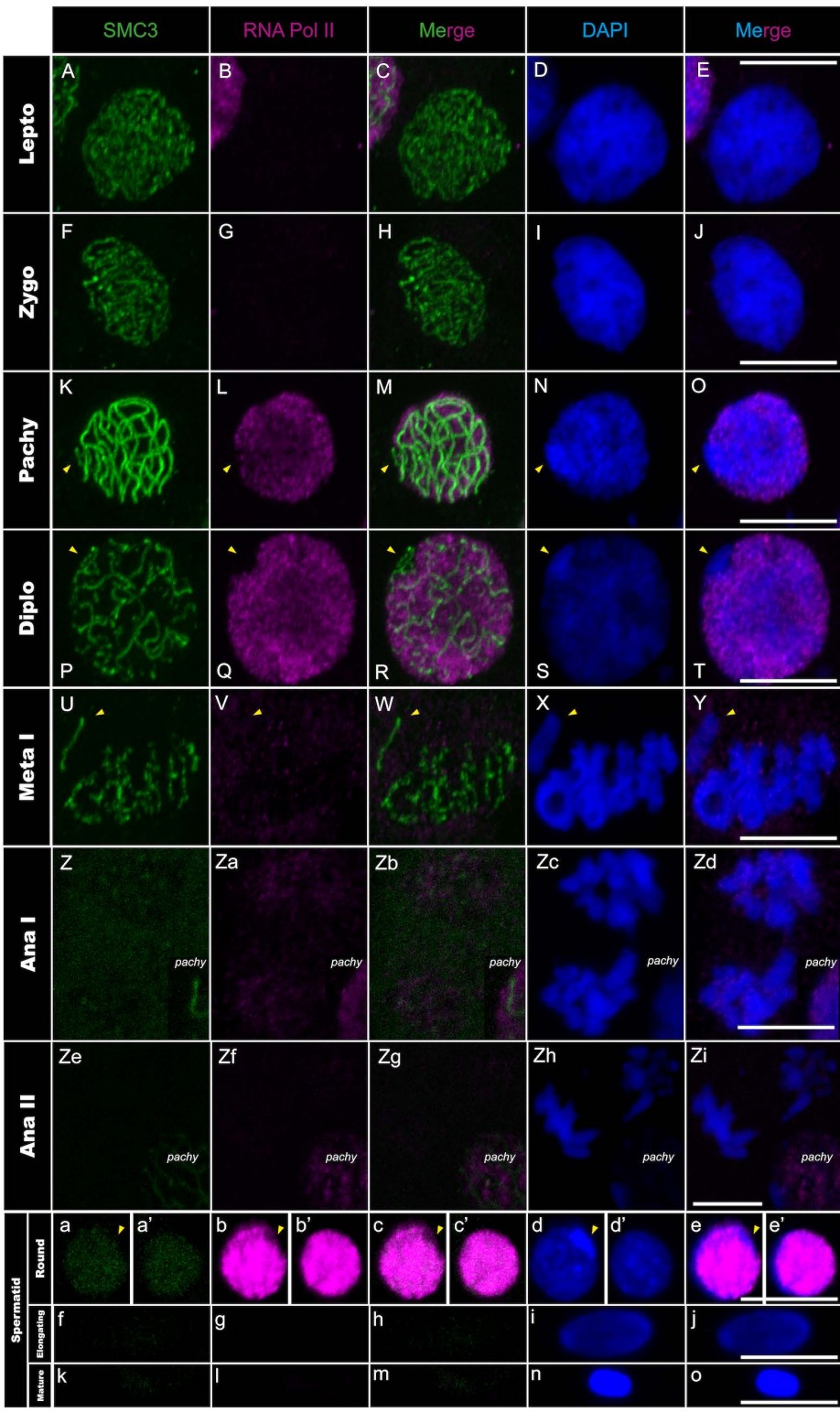

**Fig 3. RNApol-II distribution in *T. californicum* meiocytes and spermatids.** Projections of stack images acquired across *T. californicum* squashed meiocytes (A–Zi) and spermatids (a–o), at the stages indicated, stained with DAPI (blue) and double immunolabeled for SMC3 (green) and RNApol-II (magenta). RNApol-II labeling is intense from pachytene to diplotene, and in the round spermatid stage. During all spermatogenesis stages, the X chromosome is extremely reduced in RNApol-II labeling. The position of the X chromosome is indicated with arrowheads. Scale bar: 10 µm.

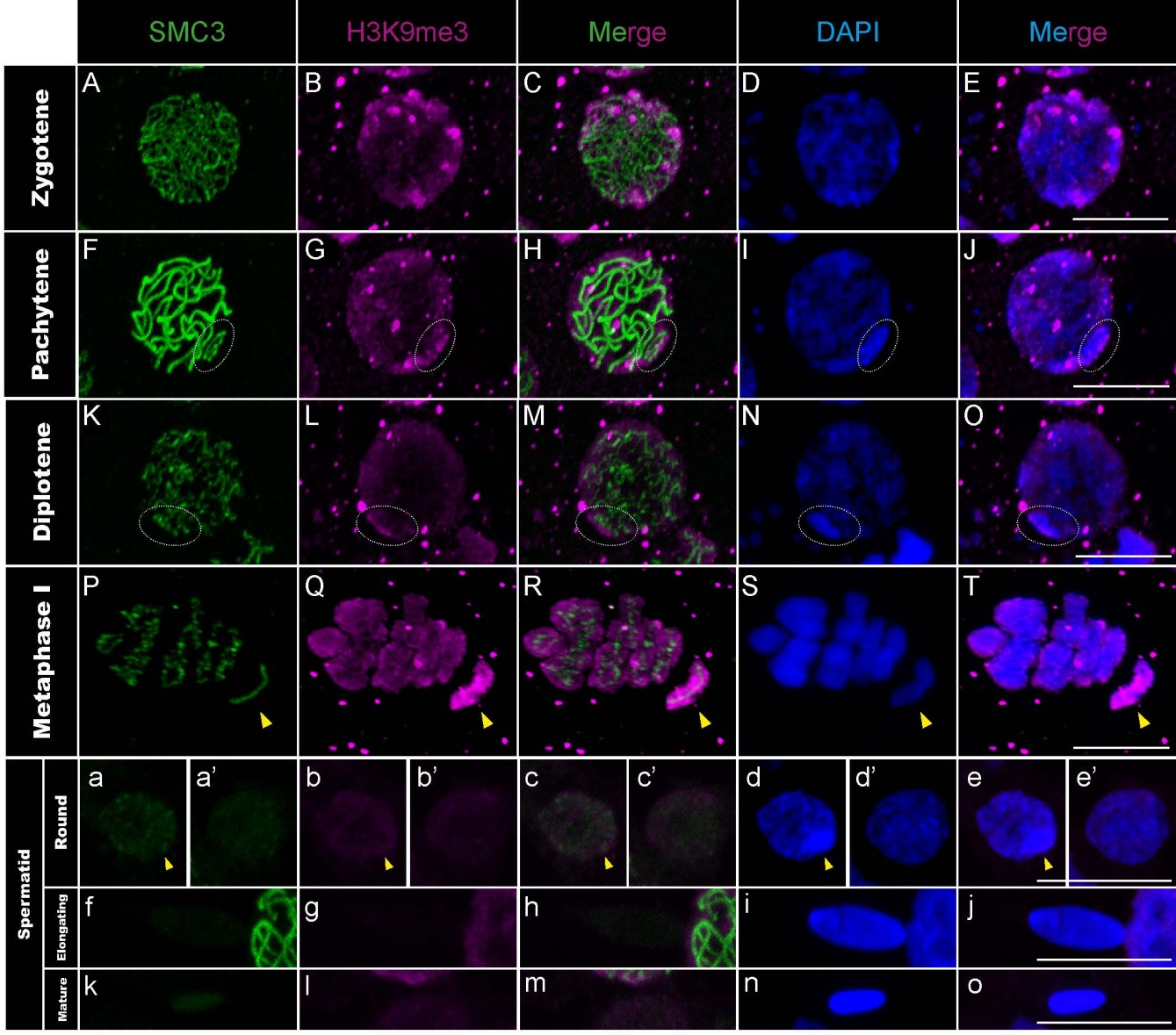

**Fig 4. H3K9me3 distribution in *T. californicum* meiocytes.** Projections of stack images acquired across *T. californicum* squashed meiocytes (A–T) and spermatids (a–o), at the stages indicated, stained with DAPI (blue) and double immunolabeled for SMC3 (green) and H3K9me3 (magenta). H3K9me3 labeling is particularly intense, coating the X chromosome from the pachytene stage. During spermiogenesis, H3K9me3 signal is absent on the X chromosome. The position of the X chromosome and the associated H3K9me3 signal is indicated with dashed circles in pachytene/diplotene and with arrowheads in metaphase I/round spermatids. Scale bar: 10 μm. See Fig D in S1 Text for screens in males of other *Timema* species.

[15,42]). However, further functional validations such as knockdown or knockout experiments will be required in the future, as currently not accessible in the system, to draw firm conclusions on the mechanisms associated with the initiation of MSCI in *Timema*. The second step is also marked by a first wave of transcription on the autosomes from pachytene to diplotene, while the X chromosome is maintained transcriptionally silenced throughout this entire period (Fig 3K–3T). In the third step, transcription is once again greatly reduced or entirely absent across all chromosomes during metaphase I. However, unlike in the first step, the X chromosome can be distinguished from the others due to the presence

of H3K9me3 modification coating its entire length (Figs 3U–3Y, 4P–T and Fig D panels D-F in S1 Text). This enrichment of H3K9me3 modification on the X chromosome, compared to the autosomes, was previously shown and quantified for *Timema* male meiosis using H3K9me3-directed ChIP-seq analyses [24]. The fourth step marks the presence of the heteropycnotic body until the round spermatid stage and the initiation of the second wave of autosomal transcription (Fig 3a–3e'). This step is also marked by some variation in the maintenance of H3K9me3 among *Timema* species (Fig 4a–4e' and Fig D panels A-C and D panels G-P in S1 Text), suggesting that this epigenetic modification is not essential for the maintenance of X silencing at this stage. Moreover, we note that H3K9me3 signals in the autosomes vary among species at this and earlier stages, with for instance the presence of intense foci in the nuclei of *T. petita* or *T. cristinae* that might indicate clusters of heterochromatin regions (Fig D panels A-C and D panels L-P in S1 Text). Finally, the last step involves the loss of MSCI and transcriptional markers in all species as spermatids elongate and mature, as evidenced by the absence of the heteropycnotic body, as well as the loss of RNA polymerase II and H3K9me3 signals (Figs 3f–3o, 4f–4o and Fig D panels D-F in S1 Text).

## Dynamics and assembly of centromere proteins during spermatogenesis

Interestingly, the first wave of transcription in *Timema* does not only coincide with the initiation of MSCI but also with a change in chromosomal distribution of the centromere histone variant CenH3, which we described previously [25]. Specifically, *Timema* is a monocentric species and its CenH3 is expected to localize to confined chromosomal regions, where it recruits other centromere proteins to ensure proper chromosome segregation during cell division [18,25]. However, we find that the absence of transcription from autosomes and the X chromosome during leptotene and zygotene in meiosis I coincides with the recruitment of CenH3 along the full length of all chromosomes (Figs 3A–3J, 5A–5J but also [25]). It is only during the transition from zygotene to pachytene that cells establish the expected monocentric distribution of CenH3 on the autosomes but not on the X, which we here show occurs in parallel to the activation of transcription from the autosomes and silencing of the X (Figs 3K–3O, 5K–5O and Fig E in S1 Text).

Considering this parallel occurrence and the functional relationships previously established between transcriptional activity and the deposition of centromere proteins in *C. elegans* and *B. mori* [45,46], we examined whether a broader correlation could exist between transcription and the recruitment of centromere proteins in *Timema*. For this, we extended our previous observations on CenH3 to include other centromere proteins and additional stages of spermatogenesis (i.e., not examined in [25]). This allowed us to integrate the recruitment dynamics of CenH3 and other centromere proteins with the progression of spermatogenesis, which has generally been poorly assessed.

We first analyzed the recruitment dynamics of CenH3 on autosomes and the X chromosome during prophase I and evaluated its conservation across *Timema* species, as our previous observations focused on a single species at this stage [25]. We found that once meiosis initiates (i.e., lepto- to zygotene stage), CenH3 is recruited over the entire length of every chromosome in all examined *Timema* species (Fig 5A–5J, Fig G panels A-I in S1 Text and see Fig F in S1 Text for CenH3 signal before meiosis), confirming that the longitudinal recruitment is evolutionarily conserved in this clade. However, the shift of CenH3 from a longitudinal distribution to a single focus on the X occurs only after the first meiotic division, not during prophase I as seen in autosomes (a detailed characterization of CenH3 dynamics on the autosomes during prophase I was previously documented in [25], but can still be visualized in Fig 5A–5T and see Fig 5U–Zd for the shift of CenH3 on the X). Moreover, this shift is not accompanied by an activation of transcription as the X remains silenced beyond the first meiotic division (Figs 3U–3Zd, 5U–5Zd). Thus, although the distribution of CenH3 and transcriptional activity are correlated on the autosomes, this is not the case for the X, where the loss of the longitudinal distribution of CenH3 from anaphase I onwards is not accompanied by a transcriptional activation (Fig E in S1 Text).

We then investigated whether the anti-correlation observed between transcription and CenH3 recruitment on autosomes during early meiosis also applies to other centromere proteins. We characterized the chromosomal distribution of two kinetochore proteins, one from the inner (CenpC) and one from the outer region (Ndc80), as well as the Bub1 protein

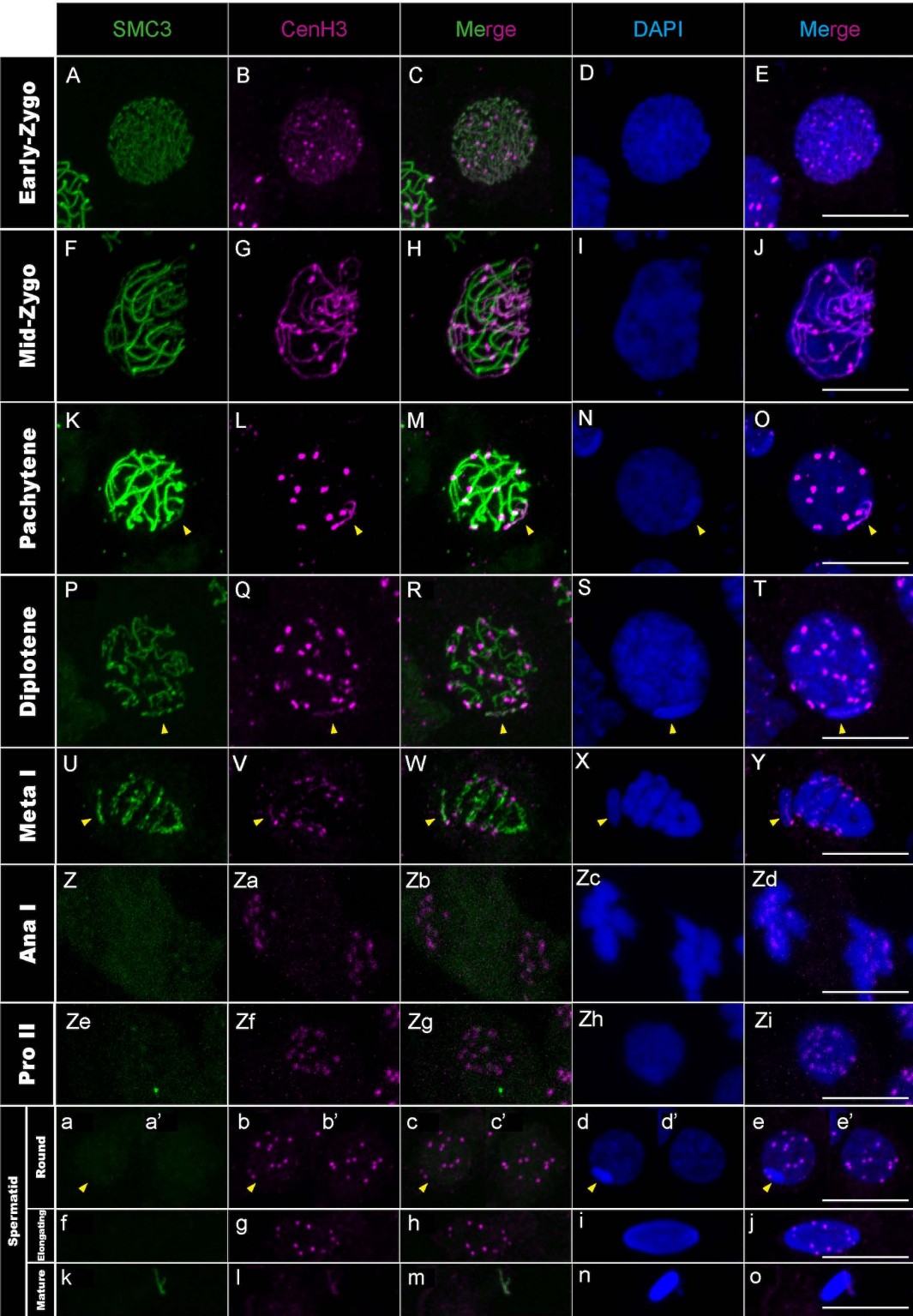

**Fig 5. CenH3 distribution in *T. californicum* meiocytes and spermatids.** Projections of stack images acquired across *T. californicum* squashed meiocytes (A–Zi) and spermatids (a–o), at the stages indicated, stained with DAPI (blue) and double immunolabeled for SMC3 (green) and CenH3 (magenta). While CenH3 is distributed along chromosomal axes at the onset of meiosis (early zygotene), it becomes restricted to foci, first in autosomes

as synapsis progresses (from mid-zygotene to metaphase I), then also in X chromosome from anaphase I. The position of the X chromosome is indicated with arrowheads. Scale bar: 10 µm. See Fig G in S1 Text for screens in males of other *Timema* species.

from the spindle assembly checkpoint (SAC) [47–49]. However, none of the three proteins are present at early prophase stages (i.e., lepto- and zygotene stages) (Figs 6A–6J, 7A–7E and 8A–8E), indicating that lack of transcription does not support the recruitment of these proteins at these stages. Instead, we observed that CenpC binds in a monocentric fashion on all chromosomes from pachytene until the end of the first meiotic division (Fig 6K–6Y and Fig H panels A-O in S1 Text), revealing that the recruitment of this inner kinetochore protein, which directly interacts with CenH3 in other organisms [19], cannot be solely attributed to CenH3 deposition during male meiosis in *Timema*. On the other hand, the outer kinetochore protein Ndc80 and SAC protein Bub1 load in a monocentric fashion on every chromosome only at pro-metaphase I, which corresponds to the stage of nuclear-envelope breakdown in several organisms (Figs 7P–7T, 8K–8T and Fig I panels P-T and J panels A-E in S1 Text). The meiotic regulation of these two proteins and associated complexes have only been studied in a handset of species [47,50], yet our findings are consistent with the idea that the assembly of the outer kinetochore region is tightly regulated in time, taking place just before segregation to prevent premature microtubule–kinetochore interactions [51].

During the second meiotic division, all four centromeric proteins are found to localize to centromeres (Figs 5Ze–5Zi, 6Z–6Zi and Fig H panels A-E, I panels F-J and J panels F-J in S1 Text). However, their timing of unloading differs. CenpC and Bub1 unload rapidly after the second meiotic division, as evidenced by the loss of signal from the round spermatids onwards (Figs 6a–6-o, 8a–8-o and Fig H panels A-E in S1 Text). By contrasts, the CenH3 signal is maintained until the elongated spermatid stage but absent from all chromosomes in mature sperm cells (Fig 5a–5o and Fig G panels A-I in S1 Text), which is either caused by an active removal of the protein as in *Drosophila virilis* and *Caenorhabditis elegans* [45,52], or because the chromatin is too compacted at this stage for antibody accessibility. Finally, Ndc80 displays a surprising cellular pattern after meiosis, accumulating in a few regions until the formation of mature sperm cells where it disappears (Fig 7a–7o). Although this cellular localization remains unexplained, it is very unlikely to reflect clustering of centromeres as they appear distinctly visible when marked with the CenH3 protein at this stage (Fig 5a–5j).

Taken together, this study represents one of the rare instances describing the cellular dynamics of essential centromere proteins during spermatogenesis, revealing that the regulation of chromosomal loading and unloading of these proteins is tightly regulated in space and time. Initially, CenH3 distributes along the entire length of all chromosomes while the other centromere proteins are absent (Figs 5A–5J, 6A–6J, 7A–7J and 8A–8J). This is then followed by focal restriction of CenH3 on each autosome and monocentric recruitment of CenpC on all chromosomes (autosomes and the X chromosome) during synapsis (Figs 5K–5O and 6K–6O). Just before the first cellular division, the last components of the kinetochore and SAC proteins are recruited in a monocentric fashion, forming the active centromeres that attach to spindle microtubules during metaphase I (Figs 7P–T, 8K–8T and Fig I panels A-E, I panels K-T and J panels A-E in S1 Text; see also [25]). Finally, by the end of the second meiotic division, kinetochore and SAC complexes disassemble, leaving only the CenH3 proteins within the centromere regions (Figs 5a–5j, 6a–6j, 7a–7j and 8a–8j).

## Discussion

The regulation of chromosome dynamics during spermatogenesis is essential for male reproductive success. Yet, the cellular processes underlying this regulation and how they evolve across species remains poorly understood. Our data offer new insights in this regard, by first revealing that chromosome synapsis in *Timema* follows the initiation of recombination. This pattern may represent the ancestral state in eukaryotes as it is present in yeast, *Arabidopsis,* jellyfish and mouse [35–38] but absent in only *D. melanogaster* and *C. elegans* [53,54]. In insects, only *Drosophila* female meiosis

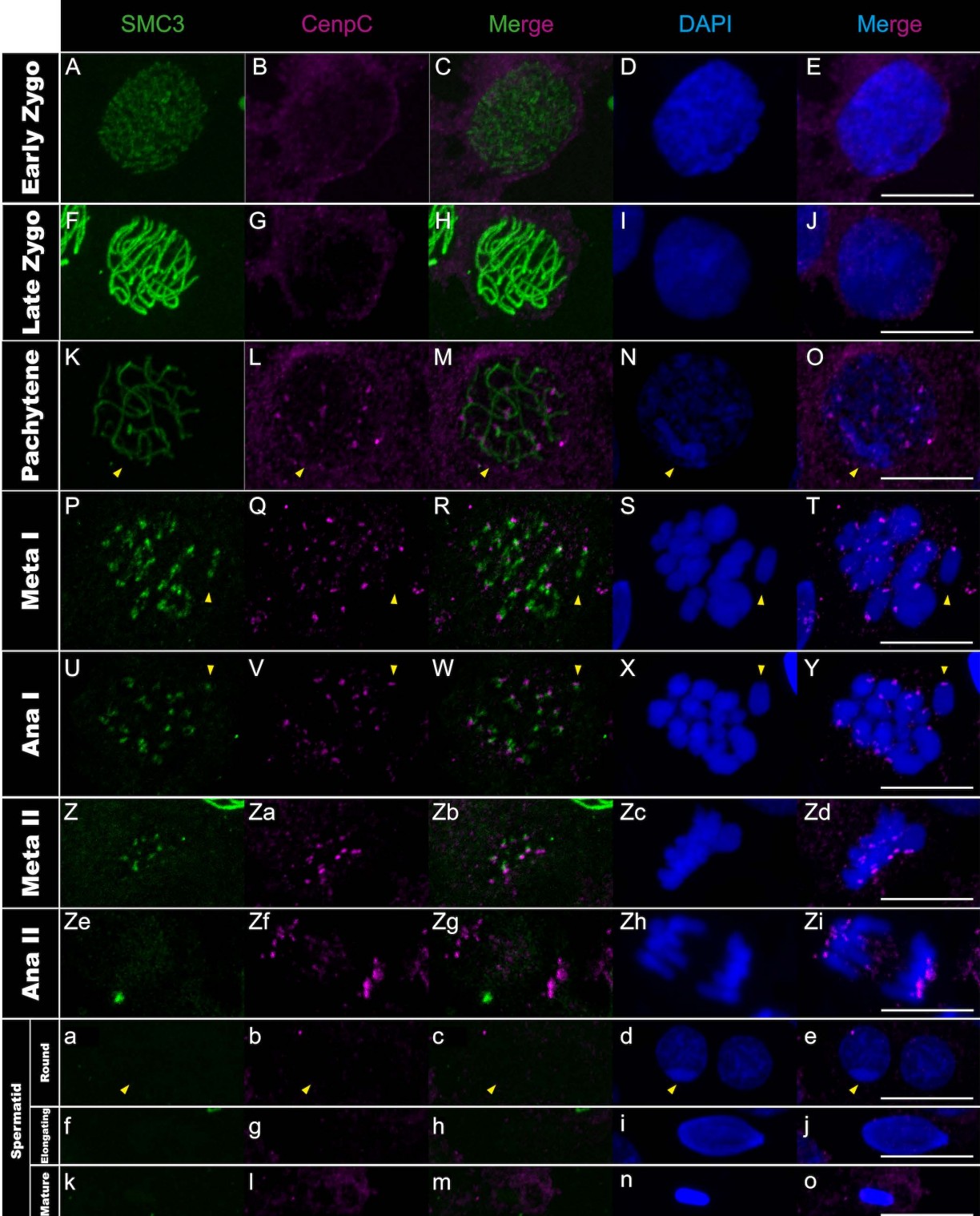

**Fig 6. CenpC distribution in *T. californicum* meiocytes and spermatids.** Projections of stack images acquired across *T. californicum* squashed meiocytes (A–Zi) and spermatids (a–o), at the stages indicated, stained with DAPI (blue) and double immunolabeled for SMC3 (green) and CenpC (magenta). The CenpC signal is not detected until the pachytene stage where it is distributed in foci on every chromosome until anaphase II. The CenpC signal is then lost at the onset of spermiogenesis. The position of the X chromosome is indicated with arrowheads. Scale bar: 10 µm.

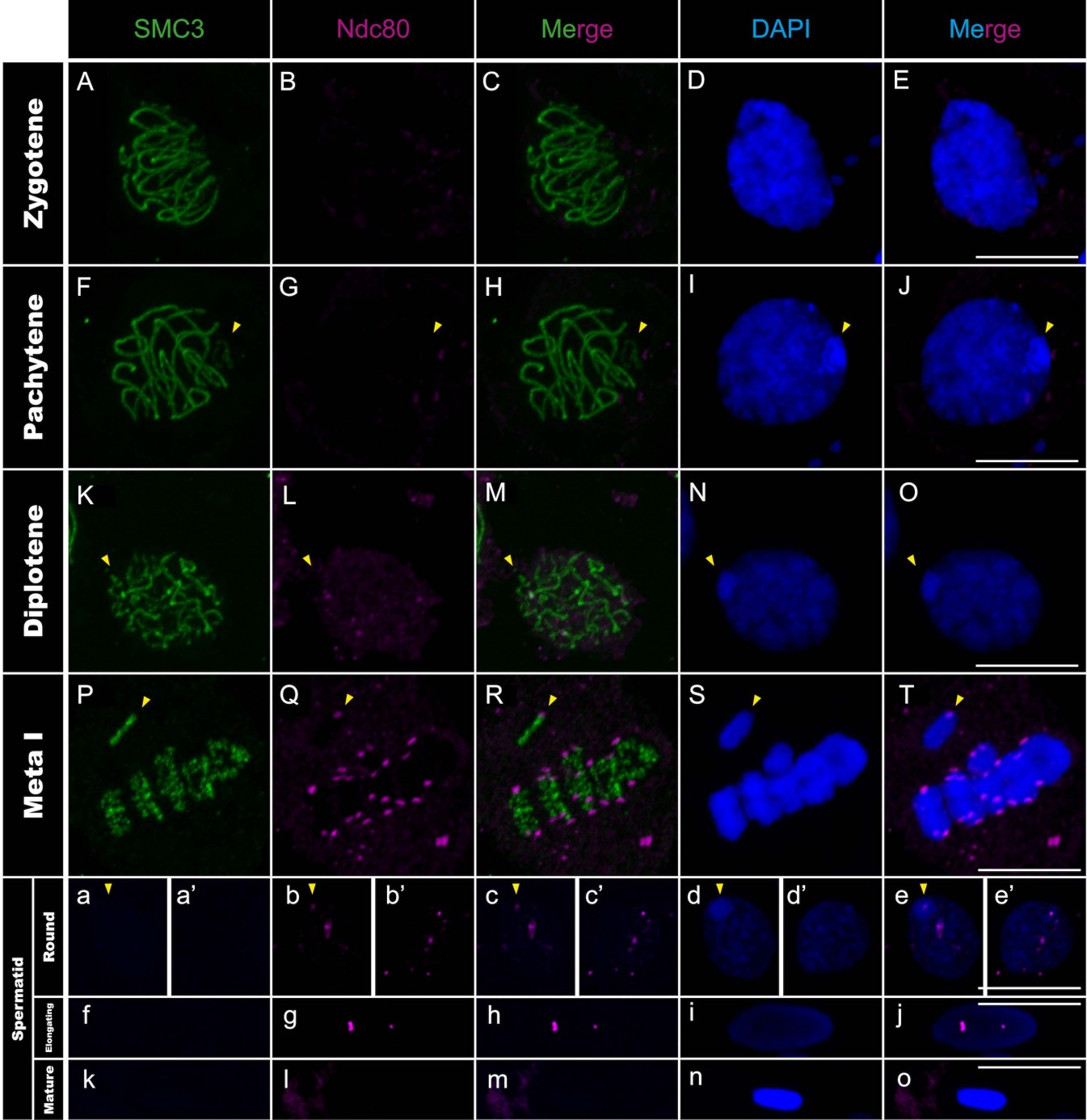

**Fig 7. Ndc80 distribution in *T. californicum* meiocytes and spermatids.** Projections of stack images acquired across *T. californicum* squashed meiocytes (A–T) and spermatids (a–o), at the stages indicated, stained with DAPI (blue) and double immunolabeled for SMC3 (green) and Ndc80 (magenta). Ndc80 signal is not detected until the pro-metaphase I stage (see Fig I in S1 Text) where it is distributed in a single focus on all chromosomes until anaphase II (see Fig I in S1 Text). The Ndc80 signal is then lost from centromeres at the onset of spermiogenesis, although two to three regions (likely non-centromeric, see main text) appear positive for Ndc80 in round and elongated spermatids. The position of the X chromosome is indicated with arrowheads. Scale bar: 10 μm.

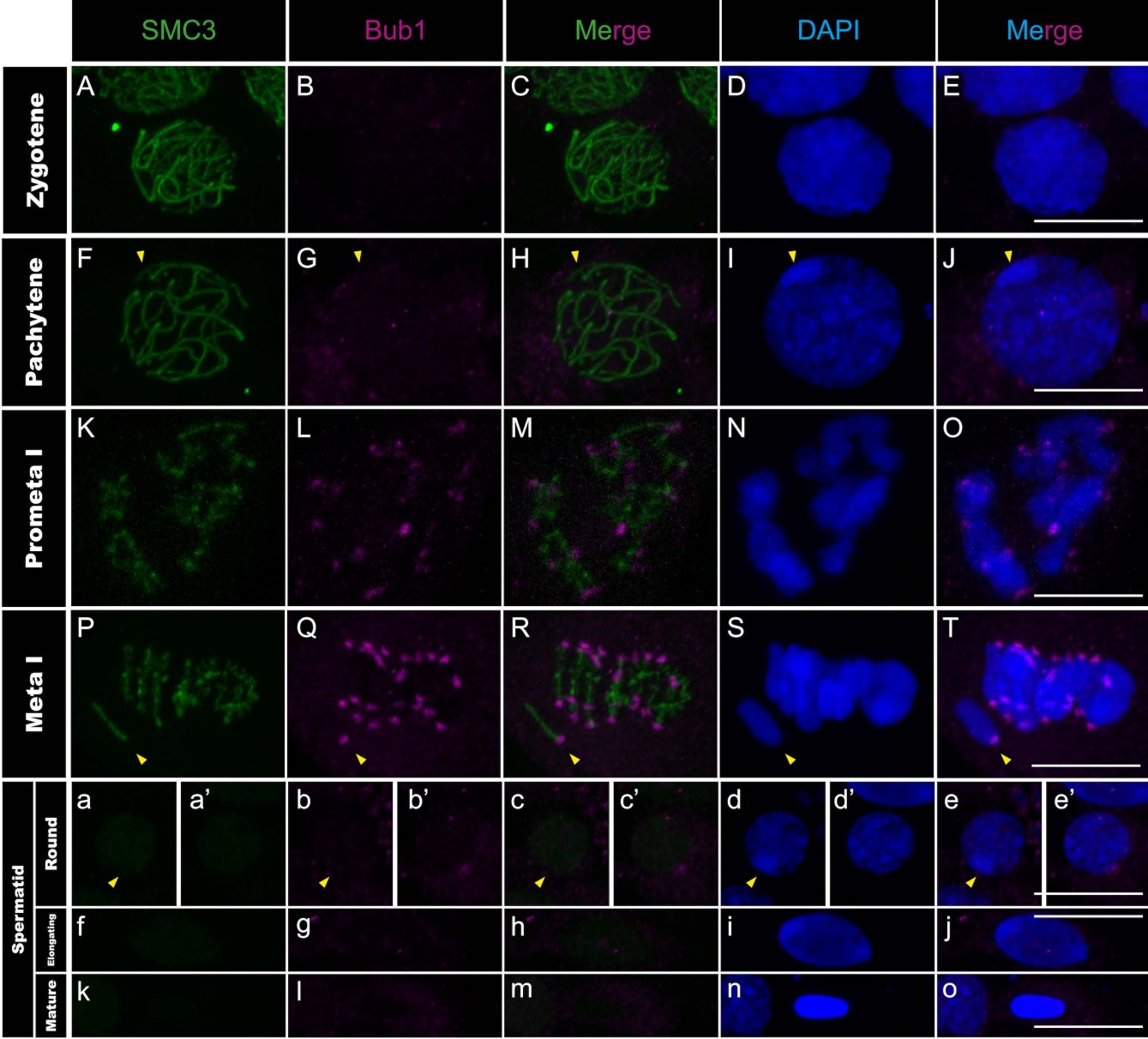

**Fig 8. Bub1 distribution in *T. californicum* meiocytes and spermatids.** Projections of stack images acquired across *T. californicum* squashed meiocytes (A–T) and spermatids (a–o), in the stages indicated, stained with DAPI (blue) and double immunolabeled for SMC3 (green) and Bub1 (magenta). Bub1 signal is not detected until the pro-metaphase I stage where it is distributed in foci on every chromosome until the second meiotic division (see Fig J in S1 Text). The Bub1 signal is then lost during spermiogenesis. The position of the X chromosome is indicated with arrowheads. Scale bar: 10 μm.

(Diptera order) is known to have synapsis preceding recombination [7,8], while two additional orders are characterized by male meiotic progressions mirroring our descriptions in *Timema* (Orthoptera: *Eyprepocnemis plorans* [9,16]; Hemiptera: *Graphosoma italicum* [33]). Nevertheless, the discrepancy in timing of recombination between *Drosophila* and other insect lineages may also originate from differences in meiotic prophase I between males and females [55,56]. Thus, further

studies on female meiosis in additional insect orders will be required before drawing firm conclusions on the evolution of recombination and synapsis in insects.

We also reveal that the lack of synapsis on the *Timema* X chromosome during the male pachytene stage is associated with the initiation of MSCI, as shown in several organisms, including mammal and other insect species [11,16]. However, the persistence of a complete transcriptional inactivity of the X chromosome throughout *Timema* spermatogenesis questions the driving forces behind the origin of the MSCI process. The evolution and key function of MSCI continues to be a source of theories and speculations, with different proximate and ultimate hypotheses proposed.

A first hypothesis posits that MSCI primarily operates to shelter unsynapsed region(s) on the sex chromosomes during the first meiotic division. Indeed, different factors generate a need for such sheltering, including damaging effects of unrepaired DSBs or ectopic recombination as well as circumventing cellular checkpoints that are active during prophase I [41,57,58]. Our findings allow us to exclude at least two such factors. First, MSCI does not appear necessary to shelter the X from the induction of DSBs during meiosis in *Timema*. Indeed, MSCI begins at a later stage than the initiation of DSBs (illustrated by the phosphorylation of H2AX in leptotene), meaning that MSCI cannot protect the X from the induction of DSBs. This contrasts with findings in other species such as the grasshopper *Eyprepocnemis plorans* (XO system) where MSCI precedes the γH2AX signal on the X [16]. Secondly, MSCI does not solely serve to circumvent checkpoints during meiosis. This is revealed by the fact that the *Timema* X chromosome maintains its condensation state and transcriptional inactivation long after the first meiotic division (i.e., until spermiogenesis and significantly past putative checkpoints), suggesting that other factors are involved in maintaining the X transcriptionally silenced.

A second, mutually non-exclusive hypothesis postulates that MSCI functions to transcriptionally silence some or all genes on the sex chromosomes that could impact cell survival during various stages of spermatogenesis and post-spermatogenesis [11,57]. Our investigation of MSCI in *Timema* supports this hypothesis, revealing that the X chromosome remains silent throughout meiosis and MSCI markers emerge only when meiotic transcription activates in pachytene. Similarly, we identified in the literature that MSCI markers are typically found in species where transcription activates during meiosis (this is notably the case in mice, opossum or grasshoppers) [14–16], while MSCI markers appear to be absent in species such as platypus (5X-5Y system) or chicken (ZW system), where global transcription is low during meiosis [12,13]. In *Timema*, we further show that a transcriptional activity post-meiosis would be sufficient to explain the maintenance of the X-silencing status until the round spermatid stage. Indeed, during this cell stage, a transcription of X-linked genes would be expected to result in major expression differences between nullo-X and X-containing spermatids. Our interpretations are again supported by findings in mice, opossum or grasshoppers, which are also characterized by transcriptional repression of sex chromosomes post-meiosis (also termed postmeiotic sex chromatin (PMSC)) [11,14,16,59].

Overall, the cell biology of MSCI in *Timema* indicates that a hypothesis based on the sheltering of unsynapsed chromosomes during meiosis I is not sufficient to fully explain MSCI and that active repression of X-linked genes throughout spermatogenesis likely plays an important role.

Finally, we show that the *Timema* X chromosome differs not only from autosomes by its transcriptional activity but also chromatin organization. Indeed, the histone variant CenH3 binds to the X along its entire length from pachytene to metaphase I, while autosomes have CenH3 only located at their centromeres during these meiotic stages (Fig 5K–5Y and Fig A in S1 Text but also [25]). These distinct distribution patterns raise important questions about the mechanisms regulating the deposition and removal of CenH3, which are known to be complex and variable among species [60].

Taken together, our findings allow us to examine three proposed mechanisms for CenH3 deposition, which involve kinetochore proteins, transcriptional activity, or SMC proteins [45,60,61]. While the first two mechanisms appear to be either incompatible or only partially applicable to *Timema*, the third mechanism aligns well with our observations. Specifically, the kinetochore-mediated recruitment of CenH3 has been identified in monocentric lineages, where specific centromere proteins—such as the kinetochore protein CenpC—play a key role in recruiting CenH3 during both mitosis and meiosis, as observed in chicken, humans, and *Drosophila* [19,62]. However, our findings exclude this hypothesis in

*Timema*: we observe CenH3 recruitment first, followed by the subsequent appearance of three other centromere proteins (i.e., CenpC, Ndc80, and Bub1), whereas this order should be reversed under a kinetochore-driven model. The transcription-based mechanism has been described in *C. elegans*, where CenH3 deposition occurs in regions with low or no transcriptional activity [45]. This low transcriptional activity (or lack thereof) may account for CenH3 deposition along all chromosomes in leptotene and its removal from autosomes in pachytene. However, it fails to explain the removal of CenH3 along the X in anaphase I (Figs 3A–3Zd, 5A–5Zd and Fig E in S1 Text). Instead, our data strongly support a third mechanism based on SMC proteins, as proposed in a recent model, where these proteins block and anchor interactions with centromeric nucleosomes during loop extrusion and yield a holocentric-like chromosome via chromatin condensation [61]. The observed dynamics of CenH3 in *Timema* male gonadal cells closely match these model predictions. While CenH3 filaments form when SMC3 proteins appear and run along the length of every chromosome during leptotene, the CenH3 filament on the X chromosome disappears when SMC3 retires from chromosomal axes in anaphase I. Interestingly, similar dynamics may also support this mechanism in certain holocentric lineages, where centromere proteins first localize between homologs, in association with axis proteins such as the cohesin complex, before relocating to more poleward-facing positions [63–65].

In addition to linking centromere protein loading with the progression of spermatogenesis, our study builds upon previous research on univalent chromosome segregation in praying mantis and grasshoppers, where kinetochore and SAC proteins are differently regulated [66–68]. In *Timema*, no differences in kinetochore or SAC proteins were identified between the univalent X chromosome and autosomes, although expanding this comparison to other centromere proteins will be necessary in the future to draw conclusions on their actual involvement.

Overall, this study highlights the importance of exploring beyond traditional model systems in cell biology to better understand the diverse processes governing male gamete formation and chromosomal inheritance. Moreover, it offers essential insights to reconstruct the evolutionary steps that have shaped this diversity.

## Materials and methods

### Timema sample collection

We used wild-collected individuals of the species *T. douglasi* (38°57'24.9"N 123°32'10.0"W), *T. knulli* (35°50'10.3"N 121°23'29.3"W), *T. petita* (36°21'32.0"N 121°54'01.3"W), *T. californicum* (37°20'35.4"N 121°38'11.3"W), *T. cristinae* (34°32'25.4"N 119°14'20.5"W), *T. monikensis* (34°06'53.6"N 118°51'11.3"W), *T. bartmani* (34°09'48.1"N 116°51'43.5"W) and *T. chumash* (34°06'53.6"N 118°51'11.3"W).

### Design, selection and validation of antibodies

A combination of custom and commercially available antibodies was employed for the immunostaining assays (Table B in S1 Text). A polyclonal rabbit anti-SMC3 antibody (Chemicon International) raised against a synthetic peptide from human SMC3 was used to detect SMC3. To detect γ-H2AX we used a monoclonal mouse antibody (Sigma-Aldrich; 05–636; Upstate) raised against amino acids 134–142 of human histone γ-H2AX [69]. This sequence has 8 identical amino acids in yeast and mouse [70]. A polyclonal rabbit anti-Rad51 antibody (Sigma-Aldrich; PC130; Oncogene Research Products), generated against recombinant HsRad51 protein, was used to detect Rad51. All three antibodies exhibited expected and similar cytological patterns to those observed in other organisms such as grasshoppers, hemipteran, *Daphnia*, or mice [9,33,35,42,43,71–73], with notably the presence of SMC3 in filaments and the appearance of an intense cloud of γ-H2AX and Rad51 foci during early prophase I (i.e., during leptotene/zygotene). Furthermore, a band at the expected size (i.e., 15kDa) could be detected by western blot for the antibody against γ-H2AX protein (Fig K in S1 Text; see [25] for details of the protocol). No band could, however, be detected for the antibody against Rad51 protein.

The presence of X heterochromatization and transcriptional activity in *Timema* male gonads has previously been tested in [24], and similarly determined in the present study, using antibodies against either H3K9me3 (#AB8898, AbCam; which labels silencing heterochromatin marks) or RNA polymerase II phosphorylated at serine 2 (p-RNApol-II (ab193468, AbCam); an indicator of transcription).

Details of the custom antibody designs for *Timema* CenH3 and kinetochore proteins can be found in [25] and in the Table B in S1 Text. For the SAC protein Bub1, we first used the protein sequence of *Drosophila melanogaster* to identify orthologs in stick insects via Protein BLAST against the NCBI database. A blast hit was already annotated as Bub1 in the stick insect *Bacillus rossius redtenbacheri* (XP_063234342.1) and was further used to identify orthologs in *Timema* via Protein BLAST against the NCBI database. The best hits of nine *Timema* species corresponded to the Bub1 ortholog as revealed by Protein BLAST against the non-redundant NCBI database. We then used the protein sequence from *T. poppense* (CAD7396045.1) to outsource to Covalab (Lyon, France) the design of three distinct peptide sequences (Table B in S1 Text). For all custom antibodies, Covalab developed polyclonal rabbit antibodies from the designed peptides, purified them via a sepharose column, and tested their specificity and immunoreactivity using ELISA assays. Moreover, all custom antibodies displayed confined signals in *Timema* male chromosomes at metaphase I, where spindle microtubules attach, consistent with the expectations for monocentric species [74].

## Immunostaining procedure with SMC3 antibody

Male gonads display a morphology resembling grapes, comprising a shoot housing mature sperm cells and an oval structure encapsulating cells at various meiotic stages (see [25]). Gonads from alive adults were dissected in 1X PBS, fixed in a solution containing 2% paraformaldehyde and 0.1% Triton X-100 for 15 minutes, followed by gentle squashing of 4–5 "grapes" onto poly-L-lysine-coated slides, then rapidly immersed in liquid nitrogen (according to [75]). After a 20-minute incubation in 1X PBS, tissues on slides were subjected to blocking with 3% BSA (in 1X PBS (w/v)) blocking buffer for a minimum of 30 minutes.

Immunostaining started by incubating slides with diluted primary antibodies in 3% BSA overnight at 4°C within a humid chamber. Exception was made for the already fluorescently labeled SMC3 antibody that was added after the following blocking step (see below). Detailed information for all antibodies used, including working concentrations, are provided in Table 2 in S1 Text. Subsequently, slides underwent three 5-minutes washes in 1X PBS, followed by a 1-hour incubation with the secondary antibody, diluted at a ratio of 1:150 in 3% BSA, at room temperature. Slides were then washed thrice for 5 minutes each, followed by a 10-minute wash in 1X PBS.

As most primary antibodies were developed from the same host (i.e., rabbit), a blocking step was performed using diluted Normal Rabbit Serum (NRS 5%) for 30 minutes at room temperature, followed by a 10-minute wash in 1X PBS. Subsequently, slides underwent a 1-hour incubation at room temperature with the already fluorescently labeled SMC3 antibody diluted at 1:100. Slides were washed thrice in 1X PBS, stained with DAPI for 3 minutes, washed one last time 5 minutes in 1X PBS, and mounted in Vectashield media.

## Immunostaining procedure without SMC3 antibody

For this procedure, no blocking steps were performed (i.e., without 3% BSA and NRS). Therefore, after a 20-minute incubation in PBS, tissues on slides were directly incubated with diluted primary antibodies in 1X PBS overnight at 4°C within a humid chamber. Exception was made for the already fluorescently labeled -tubulin antibody that was used in the following step (with the secondary antibody). Slides were then washed thrice for 5 minutes each in 1X PBS, followed by a 1-hour incubation with the secondary antibody or the already fluorescently labeled α-tubulin antibody (when needed), diluted at a ratio of 1:150 in 1X PBS, at room temperature. Slides were washed thrice in 1X PBS, stained with DAPI for 3 minutes, washed one last time 5 minutes in 1X PBS, and mounted in Vectashield media.

## Image acquisition

All acquisitions were performed using the Zeiss LSM 880 airyscan confocal microscope equipped with a 60x/oil immersion objective. All acquisitions were produced by the superimposition of focal planes. Post-processing, including cropping and pseudocoloring, was carried out using Fiji [76].

## Characterization of spermatogenetic stages and X chromosome identification

All prophase I stages could be determined based on the cellular patterns observed for SMC3. Specifically, the entry of meiosis (i.e., the leptotene stage) is characterized by numerous and thin SMC3 filaments that mark the chromosomal axes of each chromosome before homologs have paired (Fig C panel A in S1 Text; [71]). The bouquet stage, marking the entry of the zygotene stage, is evidenced by the cluster of chromosome ends forming SMC3 loops, as well as the appearance of DNA strands with DAPI staining (Fig C panels C-D in S1 Text). The progression of synapsis during zygotene is evidenced by the combination of thin and thick SMC3 filaments within the same nucleus (Fig C panels E-H in S1 Text). The pachytene stage is marked by thick SMC3 filaments, except on the X chromosome where SMC3 remains thin and the DAPI staining becomes intense (see below for details on X chromosome identification; Fig C panels I-J in S1 Text). Metaphase I and anaphase I differ by the distribution of SMC3 proteins, which stain either along chromosome axes or near centromeres, respectively (Fig C panels O-R in S1 Text). Moreover, metaphase I is characterized by a monopolar orientation of the X chromosome that does not align on the metaphase plate, contrary to autosomes (Fig C panels O-P in S1 Text).

Prophase II is characterized by round nuclei with SMC3 foci that stain again near centromeres (Fig 5Ze–5Zi). Metaphase II keeps the SMC3 foci but has no longer the X chromosome segregating as unipolar as in meta-/anaphase I (Fig C panels S-T in S1 Text), while cells in anaphase II loss the SMC3 foci and the number of DAPI-stained chromosomes is reduced (Fig C panels U-V in S1 Text).

Spermatids have several distinct features that involve the absence of SMC3 staining (Fig C panels a-f in S1 Text), a developing flagellum visible with α-tubulin staining (Fig C panels A"-C" in S1 Text), and a smaller nucleus size for the round spermatids as compared to meiotic cells (Fig C panel b and C panel b' in S1 Text).

The cytological identification of the X chromosome was previously characterized by joint analyses between cytological observations of CenH3 and H3K9me3, as well as ChIP-seq data on these same proteins [24,25]. This allowed us to further characterize the X chromosome as the chromosome with intense DAPI staining in the nucleus.

## Limitations of the study

*Timema* are non-model organisms with a one-year generation time, reaching sexual maturity at different times during the spring depending on geographical location, and with adult tissues only available for a brief period. Additionally, *Timema* are difficult to rear under laboratory conditions, and each individual produces a limited number of meiotic cells at various stages, with the majority being in the pachytene stage. These constraints limit the ability to perform consistent staining across all species and stages. However, the evolutionary dynamics of various cellular processes, within this insect group, can still be inferred by sampling a subset of *Timema* species representing the different clades ([26]; Table C in S1 Text). When inter-species variation is observed (e.g., for γH2AX and H3K9me3 signals on the X chromosome; see Tables A and C in S1 Text), its reliability was assessed by sampling multiple individuals from the same species and confirming consistent phenotypes in at least two different species.

## Supporting information

**S1 Text.** **Fig A:** γH2AX distribution at various stages of spermatogenesis for three additional *Timema* species, acquired through the projection of stack images. Focal cells at the zygotene stage are labeled with the letter *Z*, cells at the pachytene stage with the letter *P*, elongated spermatid cells with the letters ES, while yellow arrows indicate the X chromosome.

Note that at the zygotene stage, the γH2AX signal is observed at a polarized region of the nucleus, whereas at the pachytene stage, the X chromosome in *T. petita* displays γH2AX patches as in *T. californicum*, but no signal was detected in *T. cristinae* and *T. chumash*. A close-up view of the X chromosome is also shown within dashed squares for each species. Scale bar: 10 μm. **Fig B:** Rad51 distribution at various stages of spermatogenesis for three additional *Timema* species, acquired through the projection of stack images. Focal cells at the zygotene stage are labeled with the letter *Z*, cells at the pachytene stage with the letter *P*, cells in metaphase I with the letters MI, and elongated spermatid cells with the letters ES. Inset zooms on the X chromosome in pachytene are shown for *T. petita*, while in *T. chumash*, dashed circles indicate the position of the sex (X) chromosome and the associated Rad51 signal along its SMC3 axis in pachytene. Scale bar: 10 μm. **Fig C:** Characterization of spermatogenetic stages and X chromosome identification. (A-f) illustrate SMC3 and DAPI stainings throughout *T. californicum* spermatogenesis. While forming thin filaments along chromosomal axes at the onset of meiosis, SMC3 filaments appear thicker as synapsis progresses between homologs (A-J). At metaphase I, SMC3 forms cross-shaped or ring-shaped signals corresponding to rod and ring bivalents, respectively (O-P). As homologs segregate during anaphase I, SMC3 concentrates into foci near centromeres until metaphase II (Q-T). SMC3 is then lost from anaphase II onwards (U-f). Note that during the round spermatid stage, nullo-X and X-containing spermatids are represented (a-b'). (A'-L') illustrate rod and ring bivalents across 4 *Timema* species at metaphase I. Dashed squares indicate zooms of individual chromosomes illustrating rod and ring bivalents with interrupted SMC3 signal near chromosomal ends (white arrowheads in D' and G'). Note that across many metaphases I analyzed, we observed consistent formation of a single, large "ring" bivalent in *T. californicum* and *T. douglasi* but not in *T. podura* and *T. bartmani*. (A"-C") illustrate the formation of the flagellum during spermiogenesis as evidenced by α-tubulin staining in round spermatids. The position of the X chromosome is indicated with yellow arrowheads. Scale bar: 10 μm. **Fig D:** H3K9me3 distribution at various stages of spermatogenesis for three additional *Timema* species, acquired through the projection of stack images. Focal cells at the pachytene stage are labeled with the letter *P*, cells in metaphase I with the letters MI, round spermatid cells with the letters RS, and mature sperm cells with the letters MS. Note the absence of SMC3 in round spermatids and the presence of H3K9me3 coating the X chromosome in *T. petita* and *T. chumash* but not in *T. cristinae*. Arrowheads indicate the location of the X chromosome. Scale bar: 10 μm. **Fig E**: Schematic representation of CenH3 and RNApol-II distributions across different stages of spermatogenesis in *Timema*. The first activation of transcription in autosomes coinsides with the shift from a longitudinal to a focal distribution of CenH3 along these same chromosomes. However, the shift of CenH3 along the X chromosome is not associated with its transcriptional activation in anaphase I. Scale bar: 10 μm. **Fig F:** CenH3 distribution in germ cells of *T. californicum* male gonads, stained with DAPI (blue) and double immunolabeled for SMC3 (green) and CenH3 (magenta). The acquisition contrasts the distribution of CenH3 for cells in meiosis ("P" for Pachytene) and before entering meiosis ("DGC" for Diploid Germ Cell and "A" for Anaphase). Scale bar: 5 μm. **Fig G:** Projections of stack images acquired across various stages of spermatogenesis for three additional species, stained with DAPI (blue) and immunolabeled for CenH3 (magenta). Focal cells in zygotene are labeled with letter Z, cells at the pachytene stage with the letter *P*, cells in metaphase I with the letters MI, round spermatid cells with the letters RS, and mature sperm cells with the letters MS. Scale bar: 10 μm. **Fig H:** CenpC distribution at various stages of spermatogenesis for two additional species, acquired through the projection of stack images. Focal cells in leptotene are labeled with the letter L, cells at the pachytene stage with the letter *P*, cells in metaphase I with the letters MI, cells in prophase II with the letters PII, and round spermatid cells with the letters RS. Note that the lower signal intensity in the upper panel is caused by a different objective and does not reflect species variation. Scale bar: 10 μm. **Fig I:** Ndc80 distribution in various stages of spermatogenesis for two additional species, acquired through the projection of stack images. Focal cells at the pachytene stage are labeled with the letter *P*, cells in diplotene with the letter D, cells in pro-metaphase I with letters PMI, cells in metaphase I with the letters MI, cells in prophase II with the letters PII, cells in anaphase II with letters AII, cells in telophase II with letters TII, round spermatid cells with the letters RS, elongated spermatid cells with letters ES, and mature sperm cells with letters RS. Note that at the diplotene stage, chromosomes are already condensed but spindle microtubules

(immunolabeled with α-tubulin) are not yet polymerized as compared to the pro-metaphase I stage. Scale bar: 10 µm. **Fig J:** Bub1 distribution at prometaphase I and prophase II stages in *T. petita*, acquired through the projection of stack images. Scale bar: 10 µm. **Fig K:** Western blot of protein extracts from *Timema* testes, using the antibody against γH2AX. Molecular weight markers are indicated by numbers in kilodaltons (left) and their position by lines. The γH2AX antibody is expected to recognize a band around 15 kDa as indicted by the arrow. **Table A:** γH2AX distribution across species and individuals during the zygotene (bouquet) stage as well as on the X chromosome during pachytene (i.e., the initiation of MSCI). **Table B**: Antibodies and chemicals used. A combination of custom and commercially available antibodies were employed for the immunostaining assays. **Table C:** Inter- and intra-species replication of the cellular patterns described in this study as well as in [24,25]. The number of technical replicates for each individual are indicated between brackets and separated by semi-columns.
(DOCX)

## Acknowledgments

We thank current and previous members of the Schwander lab for discussions. We also thank Rocío Gómez Lencero for help in developing the immunostaining protocol in *Timema*, and Arnaud Paradis from Cell Imaging Facility (CIF, Lausanne University, Switzerland) for help with image acquisitions.

## Author contributions

**Conceptualization:** William Toubiana, Tanja Schwander.

**Data curation:** Zoé Dumas, William Toubiana.

**Formal analysis:** Zoé Dumas, William Toubiana, Marie Delattre, Tanja Schwander.

**Funding acquisition:** Tanja Schwander.

**Investigation:** Zoé Dumas, William Toubiana.

**Methodology:** Zoé Dumas.

**Supervision:** Tanja Schwander.

**Validation:** Zoé Dumas, William Toubiana, Marie Delattre, Tanja Schwander.

**Visualization:** Zoé Dumas, William Toubiana, Marie Delattre, Tanja Schwander.

**Writing – original draft:** William Toubiana, Tanja Schwander.

**Writing – review & editing:** William Toubiana, Tanja Schwander.

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
