## [Decision Letter · Decision Letter 0]

10 Jul 2025

PGENETICS-D-25-00361

Dynamics of recombination, X inactivation and centromere proteins during stick insect spermatogenesis

PLOS Genetics

Dear Dr. Toubiana,

Your revised manuscript has been evaluated by one previous reviewer and one new reviewer. I am pleased to inform you that both reviewers found the revised version to be much improved and have only minor comments. However, they make important and very helpful suggestions to improve your manuscript, and I strongly encourage you to include these changes in your final version. I will leave it to you to include or not their suggestions and I will not send the manuscript back for review. However, I will need a rebuttal letter addressing all of their comments before final acceptance of your manuscript.

Please submit your revised manuscript within 30 days Aug 09 2025 11:59PM. If you will need more time than this to complete your revisions, please reply to this message or contact the journal office at plosgenetics@plos.org. Please include the following items when submitting your revised manuscript:

We look forward to receiving your revised manuscript.

Kind regards,

Jean-René Huynh

Academic Editor

PLOS Genetics

Pablo Wappner

Section Editor

PLOS Genetics

Aimée Dudley

Editor-in-Chief

PLOS Genetics

Anne Goriely

Editor-in-Chief

PLOS Genetics

**Additional Editor Comments:**

Your revised manuscript has been evaluated by one previous reviewer and one new reviewer. I am pleased to inform you that both reviewers found the revised version to be much improved and have only minor comments. However, they make important and very helpful suggestions to improve your manuscript, and I strongly encourage you to include these changes in your final version. I will leave it to you to include or not their suggestions and I will not send the manuscript back for review. However, I will need a rebuttal letter addressing all of their comments before final acceptance of your manuscript.

**Journal Requirements:**

At this stage, the following Authors/Authors require contributions: Zoé Dumas, William Toubiana, Marie Delattre, and Tanja Schwander. Please ensure that the full contributions of each author are acknowledged in the "Add/Edit/Remove Authors" section of our submission form.

The list of CRediT author contributions may be found here: https://journals.plos.org/plosgenetics/s/authorship#loc-author-contributions

https://journals.plos.org/plosgenetics/s/submission-guidelines#loc-parts-of-a-submission

5) We notice that your supplementary Figures, and Tables are included in the manuscript file. Please remove them and upload them with the file type 'Supporting Information'. Please ensure that each Supporting Information file has a legend listed in the manuscript after the references list.

6) We note that your Data Availability Statement is currently as follows: "All the data generated in this study are available as supplementary materials.". Please confirm at this time whether or not your submission contains all raw data required to replicate the results of your study. Authors must share the “minimal data set” for their submission. PLOS defines the minimal data set to consist of the data required to replicate all study findings reported in the article, as well as related metadata and methods (https://journals.plos.org/plosone/s/data-availability#loc-minimal-data-set-definition).

7) Please provide a detailed Financial Disclosure statement. This is published with the article. It must therefore be completed in full sentences and contain the exact wording you wish to be published.

1) Please clarify all sources of financial support for your study. List the grants, grant numbers, and organizations that funded your study, including funding received from your institution. Please note that suppliers of material support, including research materials, should be recognized in the Acknowledgements section rather than in the Financial Disclosure

2) State the initials, alongside each funding source, of each author to receive each grant. For example: "This work was supported by the National Institutes of Health (####### to AM; ###### to CJ) and the National Science Foundation (###### to AM)."

3) State what role the funders took in the study. If the funders had no role in your study, please state: "The funders had no role in study design, data collection and analysis, decision to publish, or preparation of the manuscript."

4) If any authors received a salary from any of your funders, please state which authors and which funders..

8)  Please ensure that the funders and grant numbers match between the Financial Disclosure field and the Funding Information tab in your submission form. Note that the funders must be provided in the same order in both places as well.  

**Reviewers' comments:**

Reviewer's Responses to Questions

**Comments to the Authors:**

Reviewer #1: review is uploaded as attachment.

Reviewer #2: The manuscript submitted by Dumas and coworkers is an interesting study, characterizing three different aspects of meiosis in several species of stick insects belonging to genus Timema: dynamics of recombination and synapsis, inactivation of the X chromosome, and loading of centromeric proteins. The authors performed the immunolocalization of different proteins involved in these processes (SMC3, γH2AX, RAD51, H3K9me3, RNA pol-II, CenH3, Cenp-C and Bub1). Overall, the results indicate that stick insect species follow a meiotic pattern, regarding the sequence of recombination-synapsis, similar to other insects like grasshoppers and bugs. In addition, they found some differences in the dynamics of X chromosome inactivation. Finally, the most interesting and novel contribution is related to the location and dynamics of centromeric proteins, a topic that has been particularly elusive in the study of meiosis in insect.

Although the work is mainly descriptive, the comparison between different species and the inclusion of centromere analysis make it a relevant contribution. Nevertheless, there are some aspects that should be addressed by the authors (this is not an exhaustive list).

General comments:

1- the work is mainly focused on the species T. californicum. Nevertheless, the same detailed analyses are presented for almost every protein in the other species. I am not sure about why authors choose this option, but this may bias the interpretation and discussion of the results towards the patterns found in this species. It would be better if this discussion was set up from an unbiased perspective.

2- the morphological description of meiosis is too lengthy in many parts of the text. Likewise, some parts of the text seem superfluous or not properly located along the manuscript. Particularly, sections from line 227 to 262 and from line 337 to 356 could be completely eliminated or partially incorporated into the discussion section.

3- In all figures, but specially in those showing figures γH2AX, illumination of the magenta channel is too dark. I could hardly see the signals that authors describe in the text. I suggest carefully revising the levels of the images to make the signals clearly visible.

Specific comments:

4- line 68: I would better say: “Following prophase-I events, two rounds of cell…..”

5- line 86: “ Despite…..” I suggest eliminating this sentence.

6- line 120: “…recruitment of the recombinase RAD51…” I would add DMC1 as well.

7- line 160: “… formation as observed in Arabidopsis, yeast, jellyfish and mouse (35-38).” Grasshoppers and bugs, previously cited by the authors, must be added to this list.

8- line 272 and following. The point regarding the loading of CenH3 and the transcriptional activity is too confusing to me. It is not clear why authors try to make a relationship between both processes. This is perhaps related to the fact that in some species the loading of CENP-A is driven by the transcription of repetitive sequences at the centromeres. However, the discussion about this topic seems quite artificial to me. Indeed, the correlation claimed by the authors about the increase of transcriptional activity and the loading of CenH3 might be just a temporal coincidence, and not a correlation at all. Moreover, CENP-A is usually loaded at G1 stage of the cell cycle. Thus, I recommend reassessing this part of the discussion. If authors want to make a point about the relationship between meiosis progression and loading of centromeric proteins, they should revise some of the works by Suja and coworkers (PMID: 19283064, 12584241, 19283064, among others).

9- Figures are not completely comparable to each other in terms of the meiotic stages included. Some incorporate different stages of the second meiotic division, but it seems not to be a clear pattern as to why some stages are absent.

10- line 377 and following. The first sentence of this paragraph does not make any sense to me. Moreover, the following discussion about the origin of MSCI seems a bit rambling and unsupported. The fact that MSCI does not protect sex chromosomes from the production of DSBs is quite obvious from what we know in mammalian models. On the other hand, the need for silence a few genes in the X chromosome is a mere speculation in these species and is not a completely satisfactory explanation as to why the X chromosome is completely inactivated. I suggest revising the work by Viera et al., (PMID: 39766780, 34946793) for additional explanations of X chromosome inactivation in insects.

11- Discussing the configuration of centromeric proteins in holocentric chromosomes is interesting, but monocentric chromosomes of Timema species may not fit a similar pattern.

12- I believe that Rocío Gómez Ortega is incorrectly acknowledged. I suspect authors may refer to Rocío Gómez Lencero, from Universidad Autonoma of Madrid. Please, correct.

**Have all data underlying the figures and results presented in the manuscript been provided?**

Reviewer #1: Yes

Reviewer #2: Yes

PLOS authors have the option to publish the peer review history of their article (what does this mean? ). If published, this will include your full peer review and any attached files.

**Do you want your identity to be public for this peer review?** For information about this choice, including consent withdrawal, please see our Privacy Policy .

Reviewer #1: No

Reviewer #2: No

**Figure resubmission:**
---

## [Editor Report · Decision Letter 1]

1 Aug 2025

Dear Dr Toubiana,

We are pleased to inform you that your manuscript entitled "Dynamics of recombination, X inactivation and centromere proteins during stick insect spermatogenesis" has been editorially accepted for publication in PLOS Genetics. Congratulations!

Yours sincerely,

Jean-René Huynh

Academic Editor

PLOS Genetics

Pablo Wappner

Section Editor

PLOS Genetics

Aimée Dudley

Editor-in-Chief

PLOS Genetics

Anne Goriely

Editor-in-Chief

PLOS Genetics

Comments from the reviewers (if applicable):

**Data Deposition**

http://datadryad.org/submit?journalID=pgenetics&manu=PGENETICS-D-25-00361R1

**Press Queries**

---

## [Editor Report · Acceptance letter]

PGENETICS-D-25-00361R1

Dynamics of recombination, X inactivation and centromere proteins during stick insect spermatogenesis

Dear Dr Toubiana,

We are pleased to inform you that your manuscript entitled "Dynamics of recombination, X inactivation and centromere proteins during stick insect spermatogenesis" has been formally accepted for publication in PLOS Genetics! Your manuscript is now with our production department and you will be notified of the publication date in due course.

With kind regards,

Judit Kozma

PLOS Genetics

On behalf of:
